# Deep Learning-Based Knee MRI Classification for Common Peroneal Nerve Palsy with Foot Drop

**DOI:** 10.3390/biomedicines11123171

**Published:** 2023-11-28

**Authors:** Kyung Min Chung, Hyunjae Yu, Jong-Ho Kim, Jae Jun Lee, Jong-Hee Sohn, Sang-Hwa Lee, Joo Hye Sung, Sang-Won Han, Jin Seo Yang, Chulho Kim

**Affiliations:** 1Department of Neurosurgery, Hallym University College of Medicine, Chuncheon 24252, Republic of Korea; kyungckm@gmail.com; 2Division of Big Data and Artificial Intelligence, Institute of New Frontier Research, Hallym University College of Medicine, Chuncheon 24252, Republic of Korearabiting@hallym.or.kr (S.-W.H.); 3Department of Anesthesiology, Hallym University College of Medicine, Chuncheon 24252, Republic of Korea; poik99@hallym.or.kr (J.-H.K.); iloveu59@hallym.or.kr (J.J.L.); 4Department of Neurology, Hallym University College of Medicine, Chuncheon 24252, Republic of Korea; deepfoci@hallym.or.kr (J.-H.S.); bleulsh@naver.com (S.-H.L.); centertruth@naver.com (J.H.S.)

**Keywords:** foot drop, common peroneal nerve palsy, magnetic resonance image, deep learning, convolutional neural network

## Abstract

Foot drop can have a variety of causes, including the common peroneal nerve (CPN) injuries, and is often difficult to diagnose. We aimed to develop a deep learning-based algorithm that can classify foot drop with CPN injury in patients with knee MRI axial images only. In this retrospective study, we included 945 MR image data from foot drop patients confirmed with CPN injury in electrophysiologic tests (*n* = 42), and 1341 MR image data with non-traumatic knee pain (*n* = 107). Data were split into training, validation, and test datasets using a 8:1:1 ratio. We used a convolution neural network-based algorithm (EfficientNet-B5, ResNet152, VGG19) for the classification between the CPN injury group and the others. Performance of each classification algorithm used the area under the receiver operating characteristic curve (AUC). In classifying CPN MR images and non-CPN MR images, EfficientNet-B5 had the highest performance (AUC = 0.946), followed by the ResNet152 and the VGG19 algorithms. On comparison of other performance metrics including precision, recall, accuracy, and F1 score, EfficientNet-B5 had the best performance of the three algorithms. In a saliency map, the EfficientNet-B5 algorithm focused on the nerve area to detect CPN injury. In conclusion, deep learning-based analysis of knee MR images can successfully differentiate CPN injury from other etiologies in patients with foot drop.

## 1. Introduction

Foot drop is a symptom of the weakness of the dorsiflexor of the ankle, and caused by entrapment, compression, or injuries of the central and peripheral nerve distributing ankle flexor muscles [1]. Because it is a quite common condition with various etiologies, precise localization or optimal diagnosis of foot drop is a prerequisite to planning treatment and rehabilitation strategies [2]. The most common causes of foot drop are common peroneal nerve (CPN) injury, followed by lower lumbar radiculopathy, and sciatic nerve lesion [1,3]. In order to confirm foot drop due to CPN injury, abnormality in a nerve conduction study should be confirmed through electrophysiologic studies [4]. However, since these electrophysiological abnormalities usually occur 1–2 weeks after the onset of symptoms, these studies may not be appropriate as a highly sensitive test for early diagnosis of CPN injury [5]. Clinical evaluation alone may miss foot drop due to cerebral infarction or lumbosacral radiculopathy, so clinical suspicion and accurate localization via electrophysiologic studies are necessary for the differentiation of foot drop [1]. In 1992, an MR neurography technique called the T2+fat suppression sequence was introduced for the diagnosis of peripheral neuropathy [6]. These types of MRI sequences were also performed in patients with foot drop due to CPN injury, and focal peroneal nerve swelling, nerve discontinuity, denervation muscle edema, or muscle atrophy in the peroneus longus muscle were commonly accompanied by CPN injury [7]. Secondary changes due to this CPN denervation can also be observed in the extensor digitorum longus and tibialis anterior muscles of the anterior compartment of the lower leg. In addition, muscle atrophy due to denervation can progress to a state of chronic denervation as the muscle shows secondary fatty changes [8]. However, these changes can be subtle even in patients with a confirmed CPN injury upon electrophysiologic study, and can go undetected, even by a specialized neuromuscular radiologist [9,10]. In particular, muscle atrophy is a more common finding in the chronic stage of denervation and is not appropriate for early diagnosis in the acute phase [11]. Instead, signal alteration rather than atrophy of affected muscles can be found in the acute stage of nerve injury, and may be helpful to detect patients with CPN injury [12,13].

Recently, with the expanded implementation of deep learning (DL) into the medical field, AI-based radiomics is being developed for early diagnosis or more accurate diagnosis of various diseases using a variety of human images [14,15]. Existing DL-based diagnostic technologies for breast and lung cancer detection use simple x-rays [16,17]; the accuracy of DL-based clinical decision support systems has already reached or surpassed the predictive ability of radiologists in the diagnosis of various clinical conditions using brain MRI [18,19]. Deep learning can not only automatically extract complex and diverse features from medical images and learn complex patterns, it can also process large amounts of data to improve model performance. It also has the advantage of transfer learning, which allows algorithms built on models previously trained on other datasets to be used on the desired data [20]. To the best of our knowledge, we found only one DL-based study that utilized high-frequency ultrasound images to classify CPN injury, but this study only included diabetic patients [21]. To date, no studies have been reported to classify CPN injuries using MR neurography images, and there have been reports of improvement in peripheral nerve visualization using AI-based reconstruction of MR neurography [22]. Therefore, in this study, we investigated whether we could detect patients with foot drop due to CPN injury using knee MRI. The convolutional neural network (CNN) model is specialized for vision, and is known to reflect local image characteristics effectively [23]. Therefore, we additionally tried to confirm, through a saliency map, which characteristics of lesions in knee MRI images classified the CPN injury group.

## 2. Materials and Methods

### 2.1. Study Participants

This was a retrospective study using knee MRI image data. We screened the subjects with a unilateral foot drop who received a knee MRI from January 2017 to December 2021 at our hospital. The Institutional Review Board at Hallym University Chuncheon Sacred Hospital approved this study, and waived the need for informed consent because this study used only de-identified image data which were retrospectively collected (IRB No. 2023-09-011). Our study was performed in accordance with the Declaration of Helsinki. These data can be provided by the corresponding author upon reasonable request.

### 2.2. Knee MRI

All patients included were scanned with 3.0 T MRI scanners (Ingenia, Phillips, Best, The Netherland or Skyra, Siemens, Erlangen, Germany). The standard parameters of all MRI neurography sequences were the following: repetition time, 2000–8000 ms; echo time, 40–100 ms; acquisition time, 4 min; slice thickness, 4 mm; and field of view, 100 mm × 100 mm. Representative images are shown in Appendix A. Usually, conventional knee MRI images are taken from the biceps femoris muscle level to the upper 1/4 of the lower leg. In order to limit the MRI images of the case group and control group to the same area, we only used the MRI sequence from the upper meniscus level to the upper 1/4 of the lower leg as input data (Figure 1).

All screened axial MRI images were performed only on the symptomatic side, and included a variety of etiologies, such as spontaneous pain or other neurologic signs or symptoms. We included patients with foot drop as the case group. Among them, we excluded patients with foot drop with specific etiologies such as entrapment by ganglion cysts, sciatic nerve injury, lumbar or other spinal cord abnormalities. In addition, all case patients were those for whom CPN neuropathy was confirmed via conduction block of the peroneal nerve in follow-up electrophysiologic studies. Knee MRI was performed only on the affected knee, and the contralateral symptomatic knee of the patient was not established as a control group; this was limited to knee MRIs of other patients without CPN injury.

### 2.3. Algorithm Training

One image unit was used for training as an input image. In other words, after shuffling knee MR images in the case, we randomly split them into training, validation, and test datasets in an 8:1:1 ratio, and allocated the control group in the same way. All knee MR images were cropped to remove black empty spaces, then converted into a 224 × 224 size image through Z-score normalization and a resizing process. We did not use data augmentation techniques such as rotation, flip, shift, or zoom in/out during preprocessing. The parameters of the algorithm were trained based on the receiver operating characteristic (roc) value, and the loss function was binary cross entropy.

We used CNN-based VGG19, ResNet152 and EfficientNet-B5 algorithms for binary classification transfer learning of knee MRI. Briefly, the VGG is a CNN model with a relatively simple structure consisting of a CNN layer and a fully connected layer [24]. ResNet is developed to reduce the amount of computational labor by using a bottleneck structure, while allowing deeper layers to be built using skip connections [25]. EfficientNet is designed by considering both the width of the model and the resolution of the input data, which are overlooked in existing CNN models, and improves the performance of the model by finding the right balance between depth, width, and resolution [26]. Schematic architectures of these algorithms are shown in Appendix A. These algorithms have been previously successfully implemented in various open-source datasets using x-rays, histopathologic slides, and CT images [27,28,29]. During the training process, we used a random search technique for hyperparameter optimization. Batch size was initially set to 5 in our system, and we explored the validation loss and performance of each batch size, increasing or decreasing it by 1. After fixing batch size, learning rate, epoch size and *β*1 values were obtained in a similar manner.

### 2.4. Saliency Maps

A saliency map can identify which parts of an image are focused on by the classifier in each DL algorithm. Using the grad-class activation map (CAM) method, we can depict the important regions of an input image at the inference level in the form of a heatmap. To visualize this, we removed the last layer of each CNN algorithm, and added a global average pooling layer and a sigmoid layer. The analysis of grad-CAM was reviewed by a neurologic specialist.

### 2.5. Statistical Analysis

Continuous and dichotomous variables were presented as median (interquartile range) and number (percent), respectively. Comparisons of independent variables between case and control group were made using the Mann–Whitney U test or chi-square test, and the significance level was set at a *p*-value less than 0.05. The performance of each CNN-based model was evaluated using the area under the receiver operating characteristic curve (AUC). We also presented the performances of each algorithm using precision, recall, accuracy, and F1 score.

## 3. Results

A total of 2386 MR images from 149 subjects were included in this study. There were no differences in age or gender between the case and the control groups (Table 1). The average number of knee MRIs used to build the model was 16 images per patient. In the case and control groups, 17 (14–21) and 16 (13–21) knee MRI images were used per patient, respectively, with no difference between the two groups (*p*-value for Mann–Whitney U test = 0.871). One MR image from the case and control groups was randomly allocated to the training, validation, and test datasets, respectively (Table 2).

Appendix A shows the training performances in the validation dataset. The ResNet152 algorithm had the highest AUC performance, followed by the EfficientNet-B5 algorithm. In the test dataset, the EfficientNet-B5 algorithm’s performance had the highest AUC (0.946), followed by the ResNet152 and VGG19 algorithms (Figure 2). Additional parameters of performance for each algorithm in the classification of the CPN and non-CPN group are presented in Table 3. After comparing all performance parameters of each algorithm, the EfficientNet-B5 algorithm was found to have the best performance.

Grad-CAM images of the best performing algorithm, EfficientNet-B5, are shown in Figure 3, and these show axial MR neurography images and concurrent saliency maps of four patients with foot drop due to CPN injuries. In these grad-CAM images, the EfficientNet-B5 algorithm paid attention to CPN areas in knee MRIs. Moreover, this algorithm focused on the nerve region though the left tibialis anterior muscle; atrophy was prominently observed (Figure 3a,b).

## 4. Discussion

In this study, we successfully classified images of patients with CPN injuries using knee MRI only. In general, in patients with CPN injury, we can confirm that the cause of their foot drop is due to CPN injury through medical history, neurologic examination, electrophysiologic study, and other diagnostic tests. However, if the results of these tests are not helpful in differentiating patients with the CPN injury, DL classification using knee MR images, especially with the EfficientNet-B5 algorithm, can help identify patients with foot drop due to CPN injury.

There are no studies that have used deep learning to differentiate or predict knee MRI pathology. As mentioned earlier, knee MRI neurography is a useful test for identifying nerve injuries of various peripheral nerves, including CPN injuries. However, in the case of motor dysfunction, such as foot drop, it may take a certain amount of time for visual abnormalities of the nerve or secondary changes in the muscles to develop [30,31]. Because physicians’ ability to read neuromuscular MRIs can vary according to their experience, subtle MRI changes can be missed upon conventional MRI reading. Therefore, MRI-based DL methods can provide robust predictive capabilities that are independent of differences in the reader’s interpretive power or fatigability in a certain situation.

We utilized the grad-CAM image method to provide an example of where the EfficientNet-B5 algorithm focuses in its prediction. In our saliency map images, the DL machine was found to be primarily focused on the CPN area of knee MRIs. This is consistent with previous reports of MRI findings in many patients with neurological injury. In particular, the second patient in Figure 3a,b has atrophy of the tibialis anterior muscle, which is prominent in the image (arrowhead), but the DL model is still focused on the area of the peroneal nerve (arrow). Goyal et al. reported that swelling, loss of normal fascicular appearance and peripheral fat stranding occur first after peripheral nerve injury, and muscle edema or fatty changes occur later [32]. In addition, edema or enlargement of the nerve may be observed in mild peripheral nerve injuries, and disruption of the nerve fascicle and secondary changes in the peripheral nerves and muscles may be found in moderate and severe nerve injuries [33]. Therefore, the attention of our CNN algorithm on the peroneal nerve portion of the knee MRI is consistent with the usual pathophysiologic process of peripheral nerve injury. The finding that the CNN algorithm EfficientNet-B5 reflects the local characteristics of the image properly is a well-known advantage of CNN-based image models [34].

In our study, the EfficientNet-B5 algorithm performed better in classifying MR images with CPN injury compared to the ResNet152 or VGG19 algorithms. One of the reasons for this is that EfficientNet is a model considering multi-scale parameters [35]. Traditional CNN models improve the performance of the model by increasing the depth of the layers [36]. CNN models have three scaling dimensions: depth, width, and resolution, of which increasing depth alone is not enough to improve the performance of the CNN model. In addition, as depth increases, prediction weights can change to fit local minima, and CNN models are becoming more sophisticated to create ways to overcome these disadvantages [37]. Since the EfficientNet model is designed by considering not only the depth of the layers but also the width and the resolution of the input data [26], it is expected to perform better than the Resnet and VGG models tested in our study.

Our study has several limitations. First, we were not able to assess when CPN injury occurred in patients with foot drop in this study. The timing of the injury is critically associated with the neuromuscular changes seen on the knee MRI. Therefore, the timing of CPN injury and MRI scans might vary between the participants, which may require more attention in interpreting the results. Second, to compare the knee MRI-based classification performance of the CPN and non-CPN groups, we only included images from the upper meniscus level to the upper quarter of the lower leg as input images for the DL model. The biceps femoris muscle is the only muscle that is innervated by the CPN in the thigh, and can commonly accompany muscle atrophy after a CPN injury. Therefore, the performance of the model and the results of the saliency map may vary depending on the extent to which knee MRI images are included as a DL model input. Third, due to the relatively small number of case patients, we present our results in terms of image-specific performance rather than patient-specific DL model performance. Differences in the patient-specific performance of each model may be addressed in future studies with additional recruitment of foot drop patients.

## 5. Conclusions

Knee MRI based-DL methods, especially the CNN-based EfficientNet algorithm, can be a useful strategy for differentiating CPN injury patients from other patients. In the diagnosis of foot drop patients, electrophysiologic study and MRI-based DL will be able to diagnose knee pathology much more accurately.

## Figures and Tables

**Figure 1 biomedicines-11-03171-f001:**
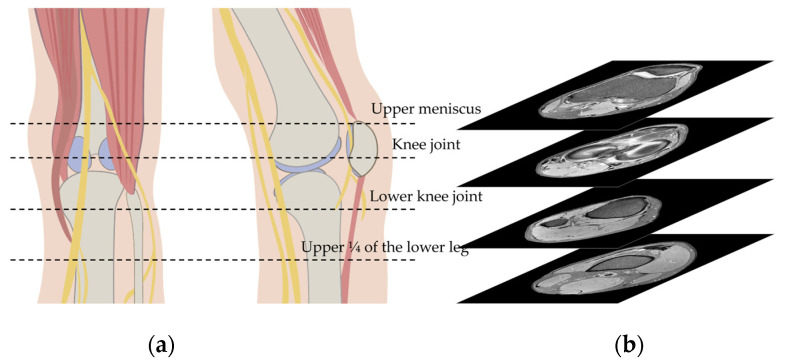
Schematic illustration of the knee (**a**) and upper and lower margin of input image in magnetic resonance images (**b**).

**Figure 2 biomedicines-11-03171-f002:**
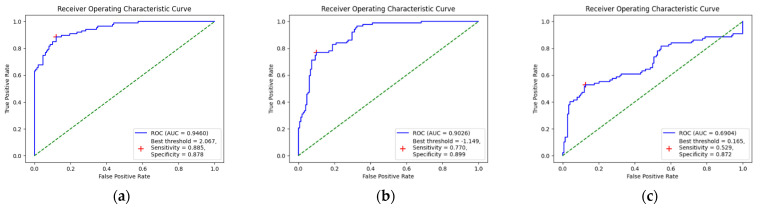
The area under the receiver operating characteristics curves of deep learning models to predict the common peroneal nerve injury magnetic resonance image in the test dataset. (**a**) EfficientNet-B5, (**b**) ResNet152, and (**c**) VGG19. AUC, area under the receiver operating characteristics curve; ROC, receiver operating characteristics.

**Figure 3 biomedicines-11-03171-f003:**
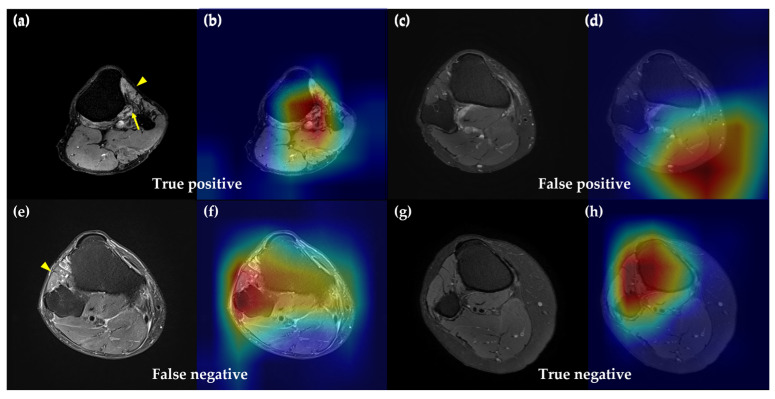
Results of grad-class activation maps showing the correct classification of knee MRI with common peroneal nerve injury. Red areas denote the areas of strongest attention. Arrows and arrowheads indicate atrophy in the tibialis anterior muscle or common peroneal nerve, respectively. True positives (**a**,**b**) and false negatives (**e**,**f**) show findings associated with common peroneal nerve injury, while false positives (**c**,**d**) and true negative (**g**,**h**) do not show similar findings.

**Table 1 biomedicines-11-03171-t001:** Baseline characteristics between the case and the control groups.

	Case Group (*n* = 42)	Control Group (*n* = 107)	*p* Value
Age, median (IQR ^1^)	51 (42–54)	53 (43–59)	0.779 ^2^
Gender, number (%)			0.927 ^3^
Male	22 (52.4)	58 (54.2)	
Female	20 (47.6)	51 (45.8)	
Lesion location			0.860 ^3^
Right	25 (59.5)	62 (57.9)	
Left	17 (40.5)	45 (42.1)	

^1^ interquartile range. ^2^ *p* for Mann–Whitney U test.^3^
*p* for χ^2^ test.

**Table 2 biomedicines-11-03171-t002:** Allocation of knee magnetic resonance images to training, validation, and test datasets.

Dataset	Case Group	Control Group	Total
Train	769	1065	1934
Validation	89	128	217
Test	87	148	235
Total	945	1341	2386

Numbers represent numbers of axial magnetic resonance images.

**Table 3 biomedicines-11-03171-t003:** Results of confusion matrix and performance metrics of EfficientNet-B5, ResNet152, and VGG19 algorithms using knee magnetic resonance images.

	TP	FP	FN	TN	Total	Precision	Recall	Accuracy	F1
EfficientNet-B5	77	18	10	130	235	0.811	0.885	0.881	0.846
ResNet152	67	16	20	132	235	0.807	0.770	0.847	0.788
VGG19	46	19	41	129	235	0.708	0.529	0.745	0.605

TP, true positive; FP, false positive; FN, false negative; TN, true negative.

## Data Availability

These data can be provided by the corresponding author upon reasonable request. The data are not publicly available due to the IRB’s decision.

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
