# Peer review of "Deep Learning-Based Knee MRI Classification for Common Peroneal Nerve Palsy with Foot Drop"

_biomedicines, 2023, doi:10.3390/biomedicines11123171_

Round 1

Reviewer 1 Report

Comments and Suggestions for Authors

This is an interesting paper that describes the feasibility of using deep learning models to detect peroneal nerve injury on MRI. Performance of three different deep learning models were compared, and additional insights were provided using saliency mapping that suggests activated areas of the model. While well-written, some areas of clarification are needed.

Introduction: 

- Clinical diagnostic performance and morphologic features of interest could be described in more detail

- Need introduction and rationale for the choice of deep learning models. 

- Past work: Are there other similar studies and what were the findings?

Methods: 

- There appears to be quite a large variation in sequence parameters? Can you provide representative images to show the variation? 

- Provide specific hyper-parameters used in the end.

- Can you include general diagram for each model highlighting the differences between models, and where the class activation branching occurs for saliency mapping?

- Table 2, can you provide number of subjects next to number of images?

Results:

- Please provide raw MRI images for positive and negative cases used for training, highlighting the areas pertinent for diagnosis, and also show saliency maps. This could provide a nice comparison between human vs. AI.

- Please also show some failure cases. this could provide insight into types of cases that may not work well with DL diagnosis.

Discussion: 

- There is no comparison to past similar studies.  There are so many studies that use AI to detect nerve damage.  How do those studies compare to yours?

Author Response

Reviewer #1.

This is an interesting paper that describes the feasibility of using deep learning models to detect peroneal nerve injury on MRI. Performance of three different deep learning models were compared, and additional insights were provided using saliency mapping that suggests activated areas of the model. While well-written, some areas of clarification are needed.

 Introduction: 

Q1: Clinical diagnostic performance and morphologic features of interest could be described in more detail

Answer1: Thank you for the comment. As the reviewer’s suggestion, we added the sentences for the detailed description of the clinical diagnostic performance and morphologic features for the CPN injury as below:

In manuscript:

However, since these electrophysiological abnormalities usually occur 1-2 weeks after the onset of symptoms, it may not be appropriate as a highly sensitive test for early diagnosis of CPN injury.[5] Clinical evaluation alone may miss foot drop due to cerebral infarction or lumbosacral radiculopathy, so clinical suspicion and accurate localization by electro-physiologic study are necessary for the differentiation of foot drop.[1] In 1992, MR neu-rography called T2+fat suppression sequence was introduced for the diagnosis of periph-eral neuropathy.[6] These types of MRI sequences were also performed in patients with foot drop due to CPN injury, and focal peroneal nerve swelling, nerve discontinuity, de-nervation muscle edema, or muscle atrophy in peroneus longus muscle were commonly accompanied by the CPN injury.[7] Secondary changes due to this CPN denervation can also be observed in the extensor digitorum longus and tibialis anterior muscles of the an-terior compartment of the lower leg. In addition, muscle atrophy due to denervation pro-gresses to a state of chronic denervation as the muscle shows secondary fatty changes.[8]

Q2: Need introduction and rationale for the choice of deep learning models. 

Answer2: Thank you for the comment. As the reviewer’s suggestion, we added some sentences for the choice of DL models as below:

In manuscript:

Recently, with the expanded implementation of deep learning (DL) into the medical field, AI-based radiomics is being developed for early diagnosis or more accurate diagnosis of various diseases using a variety of human images.[14, 15] Besides the DL-based di-agnostic technologies such as breast cancer and lung cancer detection using simple x-rays[16, 17], the accuracy of DL-based clinical decision support system has already reached or surpassed the predictive ability of radiologists in diagnosis of various clinical conditions using brain MRI.[18, 19] Deep learning can not only automatically extract complex and diverse features from medical images and learn complex patterns, but it can also process large amounts of data to improve model performance. It also has the ad-vantage of transfer learning, which allows algorithms built on models previously trained on other datasets to be used on the desired data.[20]

Q3: Past work: Are there other similar studies and what were the findings?

Answer3 : Thank you for the comment. We searched “Pubmed” with the following search terms and found no studies using MR neurography for deep learning classification. However, there was a study showing that AI-based reconstruction of MR neurography can improve visualization of peripheral nerves. We have described this further in the introduction as below:

Search terms: (peripheral nerve [TI] OR nerve [TI] OR damage [TI]) AND (deep learning [TI] OR artificial intelligence [TI]) AND (magnetic [TI] OR MR [TI])

Result of searching:

  1. Zochowski KC, Tan ET, Argentieri EC, et al. Improvement of peripheral nerve visualization using a deep learning-based MR reconstruction algorithm. Magn Reson Imaging. 2022;85:186-192.
  2. Lin J, Mou L, Yan Q, et al. Automated Segmentation of Trigeminal Nerve and Cerebrovasculature in MR-Angiography Images by Deep Learning. Front Neurosci. 2021;15:744967.
  3. Zhang L, Li Y, Bian L, Luo Q, et al. Analysis of Factors Affecting Cranial Nerve Function of Patients With Vascular Mild Cognitive Impairment Through Functional Magnetic Resonance Imaging Under Artificial Intelligence Environment. Front Public Health. 2022;9:803659.

In manuscript:

To our best knowledge, we found only one DL-based study that utilized high-frequency ultrasound image to classify the CPN injury, but this study only included diabetic pa-tients.[21] To date, no studies have been reported to classify CPN injuries using MR neu-rography images, and there has been report of improvement in peripheral nerve visuali-zation using AI-based reconstruction of MR neurography.[22] Therefore, in this study, we investigated whether we could detect foot drop patients due to the CPN injury using knee MRI. The convolutional neural network (CNN) model is specialized for vision, and is known to reflect local image characteristics effectively.[23] Therefore, we additionally tried to confirm through a saliency map which characteristics of lesions in knee MRI images classified as the CPN injury group.

Methods: 

Q4: There appears to be quite a large variation in sequence parameters? Can you provide representative images to show the variation? 

Answer4: Thank you for the comment. As the reviewer’s suggestion, we added the representative images for the MR sequence of the input data

Q5: Provide specific hyper-parameters used in the end.

Answer5: Thank you for the comment. As the review’s suggestion, actual training hyperparameters were provided in the supplemental table. (Table S1)

Table S1. Optimized hyperparameters in each CNN-based model

VGG19

ResNet152

EfficientNet-B5

Batch size

16

16

4

Learning rate

10-2

0.09

10-1

Dropout

0.1

-

0.4

Epochs

20

100

150

Q6: Can you include general diagram for each model highlighting the differences between models, and where the class activation branching occurs for saliency mapping?

Answer6: Thank you for the comment. As the reviewer’s suggestion, we added schematic diagram of the 3 CNN-based models with the saliency mapping information.

Q7: Table 2, can you provide number of subjects next to number of images?

Answer7: Thank you for the comment. Reviewer #3 had a similar question. We did not include the number of patients in Table 2 because if the seed number changes during the training process, the number of patients in the training, validation, and test datasets would be changed. If more patients with CPN injuries are recruited, the input data can be the entire MRI from one patient as one input, so these issues are described in the limitation.

In manuscript:

The average number of knee MRIs used to build the model was 16 images per patient. In the case and control groups, 17 (14-21) and 16 (13-21) knee MRI images were used per patient, respectively, with no difference between the two groups. (p-value for Mann-Whitney U test = 0.871). One MR image from the case and control groups were randomly allocated to the training, validation, and test datasets, respectively (Table 2).

Results:

Q8: Please provide raw MRI images for positive and negative cases used for training, highlighting the areas pertinent for diagnosis, and also show saliency maps. This could provide a nice comparison between human vs. AI.

- Please also show some failure cases. this could provide insight into types of cases that may not work well with DL diagnosis.

Answer8: Thank you for the comment. As the reviewer’s suggestion, we added raw and gradCAM images for the TP/FN/FP/TN cases.

Discussion: 

Q9: There is no comparison to past similar studies.  There are so many studies that use AI to detect nerve damage.  How do those studies compare to yours?

Answer9 : Thank you for the comment. We searched “Pubmed” with the following search terms and found no studies using MR neurography for deep learning classification. These sentences were added in the revised manuscript (Introduction section). Therefore, we do not describe it again in the Discussion section to avoid the redundancy. 

Reviewer 2 Report

Comments and Suggestions for Authors

Review: Deep learning-based knee MRI classification for common peroneal nerve palsy with foot drop

This paper presents a classifier for MIR foot injury determination, based on CNN pre-trained backbones, such as efficient, resent vgg19

Although the work is well organised, is not determined the novelty of the methods employed. 

I suggest the inclusion of a broader dataset regarding the same lesion/body region and explore if the models are able to generalize well in different detests. It also allows to support the discussion of the method employed and how different network architectures and schemes are sucesufull or not in the majority of the detests.

Discussion is a bit too long with no image support or results. Try to keep it shorter and support evidencing results.

Comments on the Quality of English Language

More to the point sentences. Shorter paragraphs.

Author Response

Reviewer #2.

This paper presents a classifier for MRI foot injury determination, based on CNN pre-trained backbones, such as efficient, resent vgg19

Although the work is well organized, is not determined the novelty of the methods employed. 

I suggest the inclusion of a broader dataset regarding the same lesion/body region and explore if the models are able to generalize well in different datasets. It also allows to support the discussion of the method employed and how different network architectures and schemes are successful or not in the majority of the datasets.

 : Thank you for the comment. We searched using a variety of methods and were unable to identify any public datasets that addressed CPN injury using knee MRI. However, these algorithms were working well using medical images such as x-rays, CTs, and histopathological slides in a variety of public datasets. We added this as well as the schematic depiction of the algorithms.

Discussion is a bit too long with no image support or results. Try to keep it shorter and support evidencing results.

:  Thank you for the comment. As the reviewer’s suggestion, We've tried to keep our sentences short and concise.

Reviewer 3 Report

Comments and Suggestions for Authors

This manuscript investigates deep learning (DL) classification for common peroneal nerve (CPN) injury, from knee MRI axial images. From a case group (n=42, 945 images) and control group (n=107, 1341 images), three DL algorithms were trained and validated, with EfficientNet-B5 having the best AUC=0.946 on the test set.

While a fairly straightforward study with promising results, some issues might be considered:

1. For the case group, were MRI images included only from the affected knee? For the control group, were MRI images included from both (non CPN) knees, or or one knee selected? If the latter, how was the knee selected?

2. While Figure 1 shows 4 MRI images corresponding to four key points around the knee area, 2386 MRI images from 149 subjects suggests about 16 MRI images per subject. How were these 16 MRI images selected/taken? Moreover, since 2386 is not perfectly divisible by 16, this suggests that different numbers of MRI images were taken for some subjects. This might be clarified.

3. In Section 3, it is stated that "One MR image from the case and control groups were randomly allocated to the training, validation, and test datasets, respectively". However, Table 2 shows that all 2386 MRI images were allocated to some dataset split. This statement might thus be clarified. If it means that "all MRI images from a subject were allocated to the same split, whether training, validation or test", this might be stated more clearly.

Author Response

Reviewer #3.

This manuscript investigates deep learning (DL) classification for common peroneal nerve (CPN) injury, from knee MRI axial images. From a case group (n=42, 945 images) and control group (n=107, 1341 images), three DL algorithms were trained and validated, with EfficientNet-B5 having the best AUC=0.946 on the test set.

While a fairly straightforward study with promising results, some issues might be considered:

  1. For the case group, were MRI images included only from the affected knee? For the control group, were MRI images included from both (non CPN) knees, or one knee selected? If the latter, how was the knee selected?

: Thank you for the comment. MRIs were performed only on the affected knee, and control MRIs were defined as MRIs of the knee with symptoms such as non-traumatic pain or neurologic signs that were not due to a CPN injury. In other words, we did not take an MRI of the contralateral knee of the patient's symptomatic side to serve as a control group, but limited to MRIs of the symptomatic side of other patients with non-CPN injuries. We've added additional explanation to the method for this. 

  1. While Figure 1 shows 4 MRI images corresponding to four key points around the knee area, 2386 MRI images from 149 subjects suggests about 16 MRI images per subject. How were these 16 MRI images selected/taken? Moreover, since 2386 is not perfectly divisible by 16, this suggests that different numbers of MRI images were taken for some subjects. This might be clarified.

:  Thank you for the comment. The knee MRI of a patient with CPN injury, the imaging area is wider than the conventional MRI area in the control group. We exclude non-overlapping areas in the case and control groups and include the largest possible area where changes secondary to peroneal nerve damage could be well observed. Therefore, we only included knee MRIs taken between the upper meniscus area in the proximal part and the upper quarter of the lower leg in the distal part, as shown in Figure 1. As a result, approximately16 images were used in a patient, and there was no difference in the number of knee MRIs used, with 17 (14-21) and 16 (13-21) for cases and controls, respectively. We describe this further in the Results section.

  1. In Section 3, it is stated that "One MR image from the case and control groups were randomly allocated to the training, validation, and test datasets, respectively". However, Table 2 shows that all 2386 MRI images were allocated to some dataset split. This statement might thus be clarified. If it means that "all MRI images from a subject were allocated to the same split, whether training, validation or test", this might be stated more clearly.

: Thank you for the comment. The description in method section could be confusing for readers, so we've changed it to the following.

One image unit was used for training as an input image. In other words, after shuffling the knee MR images in the case, we randomly split them into training, validation, test datasets in an 8:1:1 ratio, and allocated the control group in the same way. All knee MR images were cropped to remove black empty space, and converted it into a 224â…¹224 size image through Z-score normalization and resize process.

Round 2

Reviewer 1 Report

Comments and Suggestions for Authors

Thank you for an extensive edit. All of the concerns have been addressed.

Author Response

Your comments on this manuscript have been very helpful in improving the quality and readability of the paper. Thank you very much.

Reviewer 3 Report

Comments and Suggestions for Authors

We thank the authors for addressing our previous concerns.

Comments on the Quality of English Language

N/A

Author Response

(The authors gave the same response as above.)
